# Living with bronchial asthma: A qualitative study among patients in a hill village in Nepal

**Sabita Paudel**[1]*, **Pathiyil Ravi Shankar**[2], **Nuwadatta Subedi**[3], **Subish Palaian**[4,5]

1 Department of Pharmacology, Gandaki Medical College, Pokhara, Gandaki Province, Nepal, 2 IMU Centre for Education, International Medical University, Kuala Lumpur, Kuala Lumpur Federal Territory, Malaysia, 3 Department of Forensic Medicine, Gandaki Medical College, Pokhara, Gandaki Province, Nepal, 4 Department of Clinical Sciences, College of Pharmacy and Health Sciences, Ajman University, Ajman, United Arab Emirates, 5 Center of Medical and Bio-Allied Health Sciences Research, Ajman University, Ajman, United Arab Emirates

* drsabitapaudel@gmail.com

**Data Availability Statement:** All relevant data are within the paper and its Supporting Information files.

**Funding:** The author(s) received no specific funding for this work.

## Abstract

### Introduction

Bronchial asthma continues to be a problem in the Himalayan country of Nepal. This study explored the impact of bronchial asthma on patients' lives in a hill village in Syangja district, Nepal, and obtained information about the perceived impact of the illness, knowledge of the disease, self-care behaviors and treatment among patients.

### Material and methods

The study site is the village of Jyamire (located at an elevation between 900 to 1200 m) Syangja district. Individuals suffering from asthma residing in the village aged 18 years or above were included. Semi-structured interviews were conducted face-to-face with the respondents at their homes using an interview guide. The interviews were audio recorded, transcribed in the Nepali language, and then translated into English for further analysis.

### Results

Most participants were female, between 18 to 60 years of age, and housewives. Most houses were built of mud and poorly ventilated. Gas was used for cooking though fire-wood was also used. Most used to get an average of three serious attacks a year both during winter and summer. The themes that emerged were the number and seasonal variation in attacks, the perceived effect of asthma on their lives and social interactions, the knowledge of the interviewee about the disease, the impact of asthma on their socioeconomic status, and treatment and self-care behaviors. Residing in a hill village required them to walk up and down several times a day and the disease seriously impacted their lives. The smoke produced during different ceremonies and during cooking also worsened their asthma.

**Competing interests:** The second author, Dr Pathiyil Ravi Shankar and last author Dr. Subish Palaian are academic editors of PLoS One. The other authors have declared that no competing interests exist. This does not alter our adherence to PLOS ONE policies on sharing data and materials.

## Conclusion

Findings suggest, the existence of multiple factors, a few unique to Nepal contributing to poor asthma control. Though the recent socioeconomic improvement has led to improved prevention and treatment options, asthma seriously affected the patients.

## Introduction

Bronchial asthma is a respiratory condition where there is bronchoconstriction in response to a variety of stimuli resulting in difficulty breathing and wheezing. Asthma remains a major problem globally. The data from the last decade estimated the prevalence of the disease to be 260 million, and among these, half a million people die from the disease each year [1]. Respiratory diseases along with bronchial asthma are a major cause of morbidity and mortality in South Asian countries [2]. The prevalence of asthma is the highest in India (6.3%) among the South Asian countries [3]. In a recent systematic review, the prevalence of bronchial asthma in Nepal was found to be between 4.2–8.9% [4].

Nepal is divided into three main regions, hills, plain land, and the Himalayas (mountain). The population of Nepal is estimated at about 29.16 million [5]. People must walk across difficult terrain to access transport, educational facilities, and healthcare, particularly in the hill and Himalayan regions. A study published in 2012 mentioned insufficient medicines, distance, non-availability of staff, sickness, and money as important reasons for not being able to access healthcare facilities [6]. A survey done in Western Nepal noted that extremely cold winters, air pollution, decreasing river water quality, advancing age, use of pesticides at home, and lack of participation in awareness programs were associated with an increase in asthma prevalence [7]. They also concluded that the costs associated with the disease can further impoverish the rural population. We were not able to obtain studies on the perception of bronchial asthma patients regarding the effect of the disease on their lives.

The mid-hills have been the historical focus of development in Nepal but with rising population, land erosion, and decreased soil fertility, this region may be becoming a problem area requiring a coherent development strategy [8]. Over the last four decades, the mid-hills have lost a substantial proportion of their young population who have migrated to district headquarters, the plain Terai region, or abroad in search of better opportunities [9]. The lack of young people can affect the family structure, and the village community, and in a difficult terrain may create difficulties in accessing healthcare and treatment. A study mentioned the major problems faced by old couples ranging from anxiety, the extra burden of housework and other activities, helplessness, and a feeling of loneliness due to the migration of young people from hill villages [10]. Syangja is a hill semi-urban district located in the western part of Nepal covering an area of 1,164 km$^2$ and has a population of about 2.5 million. About 1.13 million are males and 1.49 million are females [11]. Most parts are now accessible by graveled roads. Agriculture is the main occupation.

The present study was conducted in a hill settlement (village) in the Syangja district. The success of any future interventions to reduce the asthma burden largely depends upon interventions beyond healthcare facilities and medication use and requires patients' self-involvement. The self-involvement and self-care behaviors such as medication adherence, and lifestyle modifications (dietary changes, movement uphill and downhill) can largely be influenced by the manner in which asthma people live, their environment, and other contributing factors for disease progression and treatment adherence. The perceptions of individuals

suffering from bronchial asthma about how the disease has impacted their lives and the perceived socio-economic impact are crucial for effective interventions. We were not able to locate qualitative studies assessing the impact of asthma on the well-being of affected individuals from Nepal. Hence the authors explored similar studies conducted in countries closely related to Nepal in terms of socioeconomic and other characteristics. Authors from the neighboring country, India found poor patient compliance, and parents found difficulty in managing their children's asthma [12]. Most published studies on asthma from low and middle-income countries were quantitative and explored the knowledge, attitude, practice, etc usually using a questionnaire. While these studies point to major knowledge, attitude, and practice gaps, the in-depth exploration of factors potentially contributing to disease morbidity and eventually affecting disease outcomes is lacking. The perceived impact of bronchial asthma on the sufferers' lives, social and economic situation, and challenges in accessing treatment have not been explored previously. Hence, the aim of the present study was to explore the impact of bronchial asthma on patients' lives in a hill village in Syangja district, western Nepal, and to obtain information about how patients access care and treatment and their challenges in doing so.

## Material and methods

### Study design

This was an explorative qualitative study conducted among asthma patients living in a hill district of Nepal.

### Study site

The study site is the village of Jyamire located in Putalibazaar municipality, Syangja district, Gandaki Province. The population size of the village is approximately 2000. The altitude ranges from 900 m to 1200 m. Most people of the productive age group originating from the village work as job holders in other cities of Nepal, while some are employed in foreign countries. Most people who reside in the village are farmers and a few are shopkeepers. The village is accessible by a gravel road (not pitched) and is about 4 km from the pitched highway.

The village and the adjoining areas suffer from the scarcity of water supplies. The community relies upon the water piped from a distant water source and the supply is only enough for a few hours in the morning and evening time. The people have to store water during the supply hours and frequently the supply system breaks down, so the people have to carry water from the local ponds.

### Ethical considerations

The ethical clearance was obtained from the Institutional Review Committee of Gandaki Medical College with the registration number 077/077/078 dated 03.02.2022. Each respondent participated only once in the study. Written informed consent was obtained from the respondents. There was no direct benefit for the participants. All the guidelines set by the ethical approving body have been strictly followed.

### Sampling

The sample population was chosen using a convenience sampling method to include different asthma patients according to their distance from healthcare centers, socioeconomic status, severity of asthma, etc. We chose different demographic groups to obtain a representative sample of the population of asthma patients. A convenience sampling method was chosen taking

into consideration challenges in accessing patients, incomplete documentation, the nature of the terrain, and the resources available to the authors. Within the limits mentioned, the authors wanted to include individuals from both genders, marital status, different ethnic groups and socioeconomic classes. Transcribing and analysis of data were carried out concurrently. The authors found no new themes emerging after the ninth respondent. Two additional interviews were conducted to verify this and interviews were stopped after the eleventh respondent.

### Inclusion and exclusion criteria

All individuals aged 18 years and above who were able to converse in the Nepali language and residing in the village were included. Visitors to the village and those who could not express themselves in the Nepali language were excluded. Individuals below 18 years were excluded. The village shares similarities with other hill villages and was chosen as one of the investigators hails from the village hence it was easier to establish links with local persons. Investigators contacted the local pharmacists, health assistants, and even relatives oriented to the health sector to know about the disease condition of the patient.

The concept of data saturation was used. The interviews were concluded once no new themes or subthemes emerged.

### Developing the interview guide

The guide was developed following a thorough review of the literature and discussion among the authors. The guide was pilot tested for ease of understanding and comprehensibility among a person suffering from asthma and not residing in the village. There were no problems noted and the guide was used for the interviews. The interview guide developed is shown in the S1 Appendix. The questions explored the respondent's perception regarding the impact of bronchial asthma on their lives, their socioeconomic situation, and social interactions. Self-care behaviors, respondents' knowledge about the disease, treatments used and their availability, and financial challenges faced due to the illness were studied.

### Semi-structured interviews

Semi-structured interviews were conducted face-to-face with the respondents at their homes. The interviews were done during the period from May to July 2022. The interview guide was prepared and finalized beforehand through consensus among the authors. The first author conducted all the interviews. The completed COnsolidated criteria for REporting Qualitative research (COREQ) checklist is shown as an additional file. The interviewer noted the quality of the housing, the number of persons in the household, the number of rooms, the materials used for constructing the house, the indoor ventilation, and the fuel used for cooking. Each interview lasted about 20 minutes and was audio recorded with the permission of the respondent. The interview guide was used but within limits the respondents were provided freedom to explore related topics.

### Transcription and analysis

The recorded interview was first transcribed in Nepali language and then it was translated into English by the research assistant. The transcripts were read independently by the first two authors. Important themes, subthemes, and codes were identified independently by the two individuals. They met when required to arrive at a consensus. The transcripts and the sets of

codes and themes were examined by all the researchers and differences, if any, were resolved through consensus.

The researchers used a mixture of directed and conventional content analysis to identify the main themes and areas to be explored [13]. A mixture of inductive and deductive approaches was used. The initial set of categories was developed using the interview guide. Statements and codes not fitting into these categories were grouped into new categories after discussion among the authors. Within the main categories, a conventional content analysis process was used to explore categories that emerged from the data.

## Results

A total of 11 individuals participated in the semi-structured interviews. Table 1 shows their demographic details.

Most respondents were female and housewives who also tended to their farm animals and their plot of land. Most had asthma for over seven years.

### Inspection of the house by the interviewer (first author)

The number of household members staying together ranged from one to five. Most of the houses were poorly ventilated and there were not enough windows and doors, so the smoke used to build up in the kitchen and the rooms were dark even during day time The houses were made of mud and many houses were cemented partially meaning that the floor of the rooms and verandah were cemented but the walls were made of mud and stone. Most subjects used a mixture of fuels for cooking. For cooking food for themselves, they used liquefied petroleum gas available in cylinders. Most often, for cooking food for cattle and during winters to

**Table 1. Demographic details of respondents.**

| Characteristic | Number | Percentage |
|---|---|---|
| Gender | | |
| Male | 3 | 27.3 |
| Female | 8 | 72.7 |
| Age group (years) | | |
| 18 to 59 | 7 | 63.6 |
| 60 and above | 4 | 36.4 |
| Duration of asthma (years) | | |
| <1 | 0 | 0 |
| 1–4 | 3 | 27.3 |
| 4–7 | 2 | 18.2 |
| >7 | 6 | 54.5 |
| Occupation | | |
| Not employed* | 1 | 9.1 |
| Housewife & agriculture | 8 | 72.7 |
| Student | 2 | 18.2 |
| Marital status | | |
| Married | 6 | 54.5 |
| Widow | 2 | 18.2 |
| Unmarried | 3 | 27.3 |
| Ethnicity | | |
| Brahmin | 6 | 54.5 |
| Chhetri | 4 | 36.4 |
| Thakuri | 1 | 9.1 |

* Not employed: The person did not have any land, or animals and was not employed as an agricultural laborer or in other work.

**Table 2. Themes, subthemes and quotes emerging after analysis of the interview transcripts.**

| Theme | Subtheme | Mentioned by | Quotes |
|---|---|---|---|
| Impact of the illness | Number of attacks per year and seasonal variations if any<br>Impact of asthma on daily activities, family and social interactions<br>The financial impact of the illness | Eight respondents (73%)<br>Nine respondents (82%)<br>Seven respondents (64%) | I think this happened throughout the year. But it's more difficult in winter than in summer. Foggy weather in the winter makes it more difficult. (Respondent 4)<br>Because there is no energy but people feel I am telling lie or manipulating. Sometimes, I feel energetic and at that time I go to bring grass and dry leaves for the cattle. Sometimes I return empty. My sons don't want me to rear buffaloes but I feel bad doing nothing. (Respondent 9)<br>No, I can afford it until now. It's not that difficult. When it was available, I also bought a bundle from health post. Later, they told me it was not available. (Respondent 5) |
| Respondent's knowledge of the disease | Narrowing of the airway tube | Six respondents (55%) | Mentioned in the manuscript text |
| Perceived living and socioeconomic conditions and relation to the illness | Dusty environment<br>Need to walk up and down<br>The elderly are alone as young people migrate | Six respondents (55%) | I used to have problems, but I have to survive on my own. I am alone. There is no one to take care of me. It would not be better if I eat rice only, so I eat soup, puwa. Due to this, I could walk, and move. There is no one to clean the house, no daughters-in-law, no one. (Respondent 8) |
| Treatment of the disease and self-care behaviors | How the patient treats their disease<br>Self-care behaviors that they follow | Seven respondents (64%)<br>Eight respondents (73%) | I suffered from shortness of breath so I went to Syangja hospital. Treatment there was not effective so I went to Pokhara Army Hospital, Mahendra pool. They called me after 6 months for a follow-up. Treatment at Army Hospital was also not so effective (Respondent 6)<br>It occurs once a year. It has been 2/3 years since the disease last occurred and I have been controlling it since I understood about the disease. Sometimes moderate symptoms occur. I try for self-healing for one, or two days. It usually does not get serious. If it doesn't get better then sometimes I take steam. (Respondent 2) |

get warmth, they lit firewood. Sometimes they even boiled water and cooked food using firewood. Usually, the fireplaces were outside of the house. Six subjects had domestic animals such as cows, goats, and buffaloes living in close proximity. The inspection was done only after obtaining verbal permission from the patient and the household head.

Four major themes emerged from the interviews and are shown in Table 2. These were the impact of the illness, respondents' knowledge of the illness, perceived living and socioeconomic conditions and their relation with the illness and self-care and treatment of the disease.

## The number of attacks per year and seasonal variation

This was a subtheme under the theme impact of the illness. Most used to get two or three serious attacks of asthma a year. Some had a greater number of attacks during the summer while others had a greater frequency during the winter. Most were on regular prophylactic medication/s and in some, the frequency of attacks has decreased. The region can get foggy during the winter and some respondents mentioned that they had more problems with breathing on a foggy day. A respondent mentioned,

*Although it is more in the winter months, even in the summer months I take this regularly, so it makes me comfortable to sleep but if I miss a dose for a day, and if I took it in the morning and left in the evening then it makes me difficult.* (Respondent 3)

Some respondents had a greater number of attacks with one mentioning that she had more than three or four attacks in a month.

## Impact of asthma on daily activities, family and social interactions

This was also grouped as a subtheme under the main theme of the impact of the illness. The respondents mentioned that asthma has had a moderate to severe impact on their lives. The terrain is steep, and the village is situated on the hillside at different levels. To travel from one part to the other they must climb uphill and descend downhill. Their farmlands are also situated at different levels on the hillside. The area can be dusty during the summer and foggy during the winter. Six of them have houses nearby the graveled road, and the movement of vehicles produces a lot of dust. A respondent mentioned,

> *I cannot do such things as cutting grass and firewood. The buffalo has not been reared. I cook food.* (Respondent 1)

Those who were students mentioned that they had to often miss classes due to the illness. They were occasionally hospitalized and had to be away from their friends and family. The access to the road is through rough paths along the hillside and one of the respondents mentioned that she had to be carried down by her son to the road during an attack of asthma.

Asthma significantly impacted their social interactions. The respondents live in close-knit traditional communities and people come together on different occasions ranging from planting and harvesting and different social occasions such as marriage and festivals. Their asthma worsened while harvesting paddy, and on exposure to dust and smoke. Wood is often used for cooking and women mostly do the cooking during social gatherings. The smoke worsens their asthma. Two of the respondents felt discomfort and were judged by their peers when they were not able to participate in such ceremonies. One of the respondents was condescended to by her own husband because she could not perform household work as before. They were not actively discriminated against but others felt they were not contributing their fair share to the village activities like harvesting grain and cooking food. In some cases, the illness also resulted in friction within the family. Asthma is worsened by smoke from the cooking fire and cigarette smoke. A respondent mentioned that her husband smokes and her asthma is often aggravated, A few representative quotes are shown below.

> *During sickness, there is the miss of study, which was missed a few days ago. I was trying for Japan, and there was an exam, but I had not studied, stayed at home, rested, and it became difficult to give the exam. That's it I tried for Japan but was not successful.* (Respondent 2)

> *Yes, I go but I can't do that much, I'm going to attend like if my relatives call me in the market, I must attend but can't stay for cooking which produces smoke, I do attend, but it makes me sick. That's why people know about my problem they tell me that you are a sick person, don't do it.* (Respondent 3)

> *Because there is no energy, people feel I am telling lie or manipulating others. Sometimes, I feel energetic and at that time I go to bring grass and dry leaves for the cattle. Sometimes I return empty. My sons don't want me to rear buffaloes, but I feel bad doing nothing.* (Respondent 9)

## Impact of the disease on their financial situation

The disease did have an impact on their financial situation. Most stayed in an extended family and had the support network of fellow villagers reducing the problem to a certain extent. There were no formal support networks provided by the government. The illness affected their ability to carry out their daily housework and agricultural activities. A respondent faced challenges in writing an exam to qualify for work in Japan.

The cost of the inhaler was also mentioned. The medicine was costly, but most were able to purchase it on their own or with help from their family. Three subjects presented their economic concerns with regard to purchasing the medicine as they had to purchase it throughout the year. A respondent stated,

*Each one (inhaler) cost Rs 700. (Around USD 6) Not even a month, is enough for one inhaler.* (Respondent 5)

## Respondent's knowledge of the disease

This was identified as a main theme. Most respondents did not have enough knowledge about the disease. Some were illiterate while others had only completed primary school. Only two respondents were students. These individuals had better knowledge. Most only knew the symptoms of asthma and the relieving and aggravating factors. They knew that they should continue taking the medication. The healthcare practitioner informed them depending on the severity of their condition that they should either take the medicine continuously or should take it when they feel an attack is imminent. Participants with a longer duration of illness did have better knowledge of certain aspects of the illness; younger people who were better educated had better knowledge. Those with the illness for a longer duration perceived the illness to have a greater impact on their lives.

*I do not know my disease. I don't know whether my disease is serious or moderate. When I take medicine, I am fine. If I do not take it, it gets aggravated.* (Respondent 1)

A respondent mentioned,

*That's it now to avoid the cold, to avoid smoke, to avoid the dust, that's not to play in cold. It is difficult to breathe through the chest, it is difficult to breathe through the mouth, which is why it is difficult to sleep when walking it is difficult, but after waking up, there is a slight relief* (Respondent 2)

*This tube (inside the chest) constricts when you breathe, and when it is constricted, it is difficult to breathe at night.* (Respondent 5)

## Perceived living and socioeconomic conditions of the respondents and relation to the illness

This is mentioned as a main theme though there is considerable overlap with the theme on the impact of the illness. Many mentioned that their socioeconomic condition had improved recently. The houses were ventilated though, smoke did accumulate in the kitchen. Most lived with their family. Two of the respondents seemed to be in a weaker condition than others. The locations had electricity and road access within a thirty-minute walk from the house. Most reared domestic animals and had a plot of land though the size was not determined. They perceived that their living and socioeconomic conditions were related to the disease. A respondent stated,

*The problem used to be a little bit in the previous village when going up and down, I felt like it is normal while carrying simple loads, but it increases after I came here, it becomes more difficult while I sweep the floor and made me uneasy, there is also the effect of smoke while cooking sel roti, it's like getting dizzy, it's hard to breathe* (Respondent 3)

*My hand feels burning and tingling similar to the touch of a nettle. My hand cannot grasp thgrass while cutting grass.* Respondent))

## Treatment of the disease and self-care behaviours

This was also one of the main themes identified. Respondents took treatment for the condition. Either tablets or inhalers or both were used. A healthcare practitioner (paramedic) was available in the village. The respondents purchased the medicines out of pocket and were not aware of health insurance. They obtained the medicine from the village, the nearest marketplace, or sometimes went to the state capital of Pokhara. They were aware of the importance of taking medicines regularly. Most used salbutamol orally or as an inhaler for their disease. On average, a respondent spends around Nepalese rupees (NPR) 200 (1 USD is approximately equal to 125 NPR at the time of the study) for their treatment. The cost was higher for a salbutamol inhaler at around Nepalese Rupees 250 and those using the inhaler regularly had a higher cost. A steroid inhaler was costing around NRs 700 and those on steroids for prophylaxis incurred higher costs.

*If you don't take medicine then it happens like that for one night, you should bring it immediately if the medicine will be finished tomorrow, only has for today then you should go to get it immediately.* (Respondent 3)

## Another mentioned,

*You should have inhaler medicine at your home because sometimes you won't be able to visit the hospital immediately.* (Respondent 9)

They also took other precautions like wearing a mask during activities that were dusty or while traveling.

## Discussion

The unique topography and peculiar climatic conditions necessitate asthma to be viewed with a different lens in the case of Nepal. Foggy and extremely cold weather, deteriorating river water quality high prevalence of smoking, use of firewood for cooking, indoor and outdoor air pollution, etc., make Nepal a country with a high asthma burden [7]. In the recent past, the socioeconomic status and access to healthcare facilities have improved substantially in the country leading to better healthcare indicators [14]. Though there have been studies reporting asthma burden [4, 15] and poor quality of life [16] among asthma patients, the exact human suffering among the Nepalese population due to asthma has not been reported in the literature. This qualitative research is an attempt to provide an in-depth understanding of affected individuals' perspectives on various aspects of asthma such as experiences with asthma attacks, the impact of asthma on their lives, socioeconomic status, their knowledge of the disease, treatment perspectives, and disease impact on finances among others. The findings showed significant suffering among affected people. A detailed discussion of the significant study findings is mentioned below. We were not able to locate qualitative studies conducted on this subject in Nepal and the authors compared the study findings with other studies that were often quantitative.

## Living conditions and asthma

Asthma is influenced by poor living conditions. Asthma-friendly houses (with adequate insulation, heating, or ventilation) can have a huge impact on the well-being of asthma patients [17]. As noted by the researchers, most houses were not asthma friendly and can have a negative impact on patients' health status. Another important finding is the use of cooking fuels in asthma patients' houses. Most houses used a mix of fuels (firewood and LPG). Firewood, known to have a negative health impact among asthma sufferers [18, 19] was largely used for cooking food, cattle feed, boiling water, etc. Earlier, dried animal dung (mainly buffalo) had been used as a cooking fuel in households, which was not seen during the current research. High LPG price (imported from other countries through India), and unreliable electricity system (cost, voltage issues, unavailability), makes the public rely on firewood even for boiling small quantities of water. In addition, the respondents (nearly half) had cattle (goats, cows, buffaloes) in their close proximity, another contributing factor for triggering asthma leading to poor disease control [20]. Thus, a close look at the findings showed poor living conditions that could potentially limit asthma control. Though recently a non-government organization in Nepal has introduced 'smoke-free kitchens' in a few villages and educated the public on indoor pollution and kitchen hygiene [21] more such initiatives are needed to enhance the living conditions of asthma patients for better control.

As asthma is a disease known to worsen in cold weather [22], an attempt was made to investigate the occurrence of asthma attacks and seasonal variations. Like what is already reported in the literature [23, 24] the findings showed more occurrence during winter months and foggy days. Most respondents experienced 2–3 attacks yearly and one of them even had 3–4 attacks monthly. Since asthma attack is the major cause of mortality, measures have to be ensured to educate patients on the impact of seasonal variations.

## Patient's perception of their quality of life

Asthma patients' life quality largely depends on disease [25], symptom control [26], and the perceived impact by the individual of the disease on his/her own life. A previous study in Nepal reported poor quality of life among asthma patients [15]. Similar to those previously reported in the literature, both in Nepal [15] and other countries [27–29] current findings showed a significant impact of asthma on the patients' day-to-day lives such as traveling within their locality (mainly climbing uphill), carrying household materials (in hilly terrains people are expected to carry their materials in a basket strapped to their forehead, a common practice seen in hilly Nepal), cooking (smoke being an inducer of cough and bronchoconstriction), attending classes regularly, etc. As one respondent said, at times she has to be carried by her son during an attack to reach the hospital. Thus, it is evident that asthma still has a huge impact on the day-to-day life of individuals. There may be little significant improvement in living conditions compared to the past in Nepal. In addition, asthma also has a negative impact on the social life of the respondents. Being a disease that can quickly induce symptoms, it prevented them from participating in social functions such as marriage and performing household activities which even led to some friction among family members and members of the community. In one case a sufferer's husband devalued her for not being able to perform household work. In another instance, a respondent's husband is a smoker which often led to an aggravation of her asthma attacks.

## Disease management and self-care

Proper treatment (both anti-inflammatory and rescue medications) and avoidance of triggers are the cornerstone for asthma management, which cannot be achieved without the self-

involvement of patients [30]. Self-involvement in disease management is largely dependent on the subject's knowledge of the disease and treatment. Previous quantitative studies have reported poor knowledge among asthma patients about their disease and treatment aspects including inhaler techniques [31]. The present study investigated the extent to which individuals had knowledge of the disease and treatment. Most respondents knew the symptoms, aggravating factors, and the need for continuation of medications. This information was provided by their healthcare practitioner. It was a positive effect seen as the respondents were generally aware of the use of facemasks while getting exposed to dust. Though the respondents were taking routine treatment for asthma management, the cost of medicines (both inhalers as well as others) was considered a major burden. It is worth mentioning that, the respondents however managed to buy their medications either on their own or with the help of family members. There was no availability or knowledge of insurance coverage among the respondents which could have helped them to buy the medications with less financial impact on their budget. Though some participants had partial knowledge, they were still figuring out the way out to be insured. A study conducted in the Kathmandu valley noted that the poorest quintile of households reported catastrophic household expenditure on health associated with conditions such as injuries and noncommunicable diseases like diabetes and asthma [32].

Internal migration within Nepal has led to a rapid expansion of urban areas in the low-lying Terai region and the loss of the young population in hill villages [33]. Substantial international migration is also occurring. The low number of young people (especially males) in hill villages can impact economic activities, family structure and relations, and access to treatment. Soil fertility is decreasing and land is being abandoned and there is increased growth of invasive species [34]. As mentioned by some of the respondents, remittances have also led to socioeconomic development, and households with migrants owned more land, had greater social and political awareness, and greater access to modern technology [35].

The study was among the first to examine asthma patients' perceptions regarding the impact of the disease on their lives, socioeconomic functioning, and finances in a hill village in Nepal. However, the study also had limitations. The sample selected may not be fully representative of the population though care was taken to include different subgroups of patients. Individuals below 18 years of age were not included. The authors also did not explore the impact of the recent COVID-19 pandemic on the patients' lives. The lower castes were not included in the interviews. Though at the end of each interview, a summary of what the interviewer had gathered from the interview was shared with the participants, member checking of the transcripts was not done. The study was conducted from May to July (summer months at the study site) and the possibility of seasonal variation in perceptions should be considered. We recommend that at the village level, awareness programs about bronchial asthma can be conducted both among asthma patients and the general population. The harmful effects of smoking and indoor air pollution can be highlighted. Sessions can be conducted in schools and among school children. Similar studies are required in other areas of the country.

## Conclusions

The study findings showed a negative impact of asthma on affected individuals' life in multiple dimensions such as healthy living and social interactions. The land topography, poor access to healthcare facilities in case of an attack, and poor living conditions (both indoor and outdoor) still play an important role in asthma patients' lives. Findings suggest the existence of multiple factors, a few unique to Nepal such as mountainous terrain not often accessible by a pitched road and people needing to walk uphill and downhill for daily chores, difficulty in accessing quality health care due to difficulty in access to transportation often requiring patients to be

carried by relatives. The dust resulting from vehicular movements on non-pitched roads, use of biofuel for cooking, high altitude (in many areas), and animal rearing contributed to poor asthma control. Though the recent socioeconomic improvement has led to improved treatment options, the real burden of asthma on the sufferers is still high. Considering Nepal is a country with a high prevalence of asthma, measures must be taken to contain the negative impact of asthma on the health of individuals. Interventions are needed beyond the pharmacological treatment aspects.

## Supporting information

**S1 Checklist. COREQ (COnsolidated criteria for REporting Qualitative research) checklist.**
(DOCX)

**S1 Appendix. Interview guide.**
(DOCX)

## Acknowledgments

We would like to acknowledge the study participants for allocating their time for the interview. Thanks to Mr. Bishnu Kafle, the health care worker of the village for helping us to identify the patients with asthma in the village. Mr. Giradhari Subedi, Mrs Sujata Lamichhane, and Mr. Suresh Sigdel are acknowledged for transcribing the interview recordings.

## Author Contributions

**Conceptualization:** Sabita Paudel, Pathiyil Ravi Shankar, Subish Palaian.

**Data curation:** Sabita Paudel, Pathiyil Ravi Shankar.

**Methodology:** Sabita Paudel, Pathiyil Ravi Shankar, Nuwadatta Subedi, Subish Palaian.

**Validation:** Pathiyil Ravi Shankar, Nuwadatta Subedi.

**Writing – original draft:** Sabita Paudel, Pathiyil Ravi Shankar.

**Writing – review & editing:** Pathiyil Ravi Shankar, Nuwadatta Subedi, Subish Palaian.

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
