## [Decision Letter · Decision Letter 0]

19 May 2023

PONE-D-23-12615Living with bronchial asthma: a qualitative study among patients in a hill village in NepalPLOS ONE

Dear Dr. Paudel,

Thank you for submitting your manuscript to PLOS ONE. After careful consideration, we feel that it has merit but does not fully meet PLOS ONE’s publication criteria as it currently stands. Therefore, we invite you to submit a revised version of the manuscript that addresses the points raised during the review process. Please adhere to the comments given by two reviewers.  Please submit your revised manuscript by Jul 03 2023 11:59PM. If you will need more time than this to complete your revisions, please reply to this message or contact the journal office at plosone@plos.org. Please include the following items when submitting your revised manuscript:A rebuttal letter that responds to each point raised by the academic editor and reviewer(s). You should upload this letter as a separate file labeled 'Response to Reviewers'.A marked-up copy of your manuscript that highlights changes made to the original version. You should upload this as a separate file labeled 'Revised Manuscript with Track Changes'.An unmarked version of your revised paper without tracked changes. You should upload this as a separate file labeled 'Manuscript'.

We look forward to receiving your revised manuscript.

Kind regards,

Sumal Nandasena

Academic Editor

PLOS ONE

Journal Requirements:

" Dr Shankar is an academic editor for PLoS One.  The other authors have declared that no competing interests exist."

Reviewers' comments:

Reviewer's Responses to Questions

**Comments to the Author**

1. Is the manuscript technically sound, and do the data support the conclusions?

Reviewer #1: Yes

Reviewer #2: Partly

2. Has the statistical analysis been performed appropriately and rigorously? 

Reviewer #1: N/A

Reviewer #2: N/A

3. Have the authors made all data underlying the findings in their manuscript fully available?

Reviewer #1: Yes

Reviewer #2: No

4. Is the manuscript presented in an intelligible fashion and written in standard English?

Reviewer #1: Yes

Reviewer #2: No

5. Review Comments to the Author

Reviewer #1: Paudel S, et al. present a manuscript titled "Living with bronchial asthma: a qualitative study among patients in a hill village in Nepal." Semi-structured interviews were conducted with 11 participants living with asthma. The author found that participants showed limited levels of disease knowledge, and asthma impacted their daily lives and financial circumstances. The manuscript can help design appropriate health interventions to improve asthma management in Nepal. Below are my comments by section:

Abstract:

In the result section, please match the themes provided in the main body's result section.

Introduction:

Studies to learn about asthma and its impact on quality of life have been studied in low- and middle-income countries, including Nepal. The authors should discuss existing literature and provide a more convincing reason for conducting this study.

Please rewrite this sentence "The prevalence of the disease was estimated to be 260 million according to 57 data from the last decade and half a million people die from the disease each year." (Lines 56-57)

Methods:

Please describe in detail the reason for using convenience sampling, the number of questions asked, and the description of key questions to give readers an understanding of the methodology employed.

The author can shorten the information on the study site by including only the relevant information to the study.

The authors mentioned that the population was grouped according to characteristics like age group, gender, marital status, ethnicity, and socioeconomic status and selected participants from each group. Please provide the total number of eligible participants, the number of participants approached, and the number of participants who withdrew from the study.

The author mentions the conventional and directed approaches for identifying themes and codes while also states using the inductive and deductive approaches. Please clarify which methods were used at what stage of analysis.

Results:

The author needs to organize the result section. The author should clearly state how many themes emerged during the analysis before going through each theme in detail. If possible, the author should provide a table indicating major themes, subthemes, major quotes, and the frequency of responses for each sub-themes or code.

It is unclear what self-care behavior participants followed to manage or control their asthma and the impacts of asthma on their financial situation. Please elaborate with quotes.

In Table 1, both age groups include the age 60 years. Please rectify. Also, please include other characteristics, such as marital status, ethnicity, and socioeconomic status, which the author used to select interview participants.

Line 209 through 230 is unreadable. Please correct the formatting issue.

It would be interesting to learn the difference in the knowledge and the impact of asthma among patients with different characteristics, such as participants with asthma >7 years vs. < 7 years. For example, the authors should mention whether there was a difference in asthma knowledge and the impact on daily living and financial conditions.

Discussion:

The discussion section is well-written, but the author needs to compare the findings of the study with existing literature from other low-and middle-income countries and discuss any unique results that emerged from this study that would contribute to the literature.

Conclusion:

The following sentence is vague: "Findings suggest the existence of multiple factors, a few unique to Nepal and a few other general well-known factors contributing to poor asthma control." Please state what findings are unique to Nepal based on the study findings.

The author mentions poor access to healthcare facilities as a barrier to asthma management. However, access to health care has not been clearly explained in the result section. The author should only include the conclusion supported by the study findings.

Miscellaneous Comments:

Please correct grammars, spelling, and typos throughout the document, where applicable.

Reviewer #2: L63. I doubt the population figures mentioned in the manuscript are correct. Referee 2021 Nepal population census report.

L73-77: What is the relevance to this study?

L78-79: Are these population figures correct?

Is this district in an urban setting or a rural setting?

L84-85: Can you please elaborate on the idea you are trying to express in this sentence?

L100-107: Authors can further summarize this section.

L135-L137 This sentence may be used as the opening sentence of the paragraph.

Table 1:

On what basis the duration of asthma was classified?

How you/ What is your definition of "Not employed"

L177: What methodology is used to classify a household that is not well-ventilated? What is the meaning of cemented partially?

L185: What criteria were used to identify the seriousness of the asthma condition?

L194-195: What was the evidence to back the statement?

L209-L230: Remove

The article quality may be improved a lot more with the help of an English language expert.

The authors specifically mentioned that there are restrictions on the accessibility of research data.

6. PLOS authors have the option to publish the peer review history of their article (what does this mean?). If published, this will include your full peer review and any attached files.

Reviewer #1: **Yes: **Sachita Shrestha

Reviewer #2: **Yes: **C S Edirisuriya

---

## [Author Response · Author response to Decision Letter 0]

3 Aug 2023

Pokhara, Nepal

30th July 2023

Dear Editor,

The authors of this manuscript would like to thank you for sending us the reviewers’ comments. We have taken all the measures to address the comments raised. 

The response to the comments follows:

Reviewer #1 comments: 

Paudel S. et al. present a manuscript titled "Living with Bronchial Asthma: a qualitative study among patients in a hill village in Nepal." Semi-structured interviews were conducted with 11 participants living with asthma. The author found that participants showed limited levels of disease knowledge, and asthma impacted their daily lives and financial circumstances. The manuscript can help design appropriate health interventions to improve asthma management in Nepal. Below are my comments by section:

Abstract:

In the result section, please match the themes provided in the main body's result section.

Response to the reviewer: As per the suggestion, the themes are matched with the main body’s result section in line numbers 44-47.

Introduction:

Studies to learn about asthma and its impact on quality of life have been studied in low- and middle-income countries, including Nepal. The authors should discuss existing literature and provide a more convincing reason for conducting this study.

Response to the reviewer: The authors have added a description of this towards the end of the Introduction section to justify the rationale of the study in line numbers 108-122.

Please rewrite this sentence "The prevalence of the disease was estimated to be 260 million according to 57 data from the last decade and half a million people die from the disease each year." (Lines 56-57)

Response to the reviewer: The sentence is rewritten as ‘The data from the last decade estimated the prevalence of disease to be 260 million, and among these half a million people die from the disease each year’ in the introduction section, line numbers 62-63.

Methods:

Please describe in detail the reason for using convenience sampling, the number of questions asked, and the description of key questions to give readers an understanding of the methodology employed.

Response to the reviewer: Convenience sampling was chosen because it was difficult to access data from all the patients in the village. One of the reasons is inadequate documentation and the other reason is accessibility. Logistic and financial limitations did not allow us to reach all the patients. (line 168-172)

About the key questions, detail is given in the method section, subsection ‘Developing the interview guide’, line numbers 193-200. The questions have been briefly highlighted in the Methods section and the interview guide used is shown in the appendix. 

The author can shorten the information on the study site by including only the relevant information to the study.

Response to the reviewer: As per the comment, the information is shortened in between line number 137.

The authors mentioned that the population was grouped according to characteristics like age group, gender, marital status, ethnicity, and socioeconomic status and selected participants from each group. Please provide the total number of eligible participants, the number of participants approached, and the number of participants who withdrew from the study. 

Response to the reviewer: The authors would like to thank the reviewer for this comment. However, as mentioned in the Limitations (towards the end of the Discussion section), this research did not enroll a representative population. However, it is worth mentioning that the authors used a combination of convenience and purposive sampling methods, in a setting like the present study is valuable. Based on the reviewer's suggestion, the authors have modified the ‘sampling’ section of the revised manuscript in line numbers 168-172.

The author mentions the conventional and directed approaches for identifying themes and codes while also states using the inductive and deductive approaches. Please clarify which methods were used at what stage of analysis. 

This has been clarified in the Methods section, line numbers 227-229..

Results:

The author needs to organize the result section. The author should clearly state how many themes emerged during the analysis before going through each theme in detail. If possible, the author should provide a table indicating major themes, subthemes, major quotes, and the frequency of responses for each sub-themes or code. 

Response to the reviewer: A subsection of ‘ethnicity’ has been added in Table 1, line number 235. A new table (table 2) has been added in line number 258. The results section has been modified and reorganized in line numbers 261 to 438.

It is unclear what self-care behavior participants followed to manage or control their asthma and the impacts of asthma on their financial situation. Please elaborate with quotes. 

This has been done. The impact of the disease on the financial situation has been added as a new subtheme in line numbers 331-342. Treatment of the disease and self-care is a new theme added in line numbers 409-438. There is a degree of overlap between the different themes. 

In Table 1, both age groups include the age 60 years. Please rectify.

Response to the reviewer: As per the comment, the text is identified and rectified in Table 1, line number 235.

Also, please include other characteristics, such as marital status, ethnicity, and socioeconomic status, which the author used to select interview participants. The marital status and ethnicity details have been added at the end of Table 1.

Line 209 through 230 is unreadable. Please correct the formatting issue.

Response to the reviewer: The issue is noted as per the comment and corrected.

It would be interesting to learn the difference in the knowledge and the impact of asthma among patients with different characteristics, such as participants with asthma >7 years vs. < 7 years. For example, the authors should mention whether there was a difference in asthma knowledge and the impact on daily living and financial conditions. 

While participants with a lower duration of illness did have better knowledge of certain aspects of the illness younger people who were better educated had better knowledge. Those with the illness for a longer duration perceived the illness to have a greater impact on their lives. This has been mentioned in the Results section, line numbers 351-374. (There is an issue with line numbering here that we could not correct). 

Discussion:

The discussion section is well-written, but the author needs to compare the findings of the study with existing literature from other low-and middle-income countries and discuss any unique results that emerged from this study that would contribute to the literature. 

Response to the reviewer: The authors have made a few changes in the discussion as suggested by the reviewer. However, as mentioned in the last part of the Introduction section, there have been no qualitative studies comparable to the present study. The authors have also added a statement on the lack of similar studies toward the end of the first paragraph of the ‘Discussion section’ in line numbers 455-457.

Conclusion:

The following sentence is vague: "Findings suggest the existence of multiple factors, a few unique to Nepal (high altitude, biofuel, cold weather, animal rear, graveled road, water source at distant) and a few other general well-known factors contributing to poor asthma control." Please state what findings are unique to Nepal based on the study findings.

Response to the reviewer: As per the suggestion, the text is modified as ‘Findings suggest the existence of multiple factors, a few unique to Nepal such as mountain terrain not accessible by a pitched road and people needing to walk uphill and downhill for daily chores, difficulty in accessing quality health care due to difficulty in access to transportation requiring patients to be carried by relatives. The dust resulting from vehicular movements on non-pitched roads, use of biofuel for cooking, high altitude, and animal rearing contributed to poor asthma control.’ in line numbers 575-578..

The author mentions poor access to healthcare facilities as a barrier to asthma management. However, access to health care has not been clearly explained in the result section. The author should only include the conclusion supported by the study findings. 

Response to the reviewer: The sentence is modified as suggested in the conclusion section.

Miscellaneous Comments:

Please correct grammars, spelling, and typos throughout the document, where applicable.

This has been done.

Reviewer #2: L63. I doubt the population figures mentioned in the manuscript are correct. Referee 2021 Nepal population census report.

Response to the reviewer: The issue is noted and corrected to ‘The population of Nepal is estimated at about 29.16 million’ in the introduction section, line number 70.

L73-77: What is the relevance to this study? 

The relevance has been highlighted in the introduction section line numbers 92-96.

L78-79: Are these population figures correct? 

Yes, it has been checked and verified.

Is this district in an urban setting or a rural setting?

Response to the reviewer: The district is a semi-urban one which is added in the introduction section, line number 96 and 97.

L84-85: Can you please elaborate on the idea you are trying to express in this sentence? 

The sentence has been elaborated in the introduction section.

L100-107: Authors can further summarize this section.

Response to the reviewer: The sentence has been condensed by removing some parts in method section, line numbers 132-137.

L135-L137 This sentence may be used as the opening sentence of the paragraph.

Response to the reviewer: The sentence has been reorganized in method section.

Table 1:

On what basis the duration of asthma was classified? 

There was no specific basis for the classification. We believed the groups according to the years chosen by us would represent the short, medium, and long-term duration of illness. 

How you/ What is your definition of "Not employed" 

There was one respondent mentioned in this category. The person did not have any land, or animals and was not employed as an agricultural laborer or in other work. This has been mentioned as a footnote to Table 1, line numbers 237 and 238. 

L177: What methodology is used to classify a household that is not well-ventilated? What is the meaning of cemented partially?

Response to the reviewer: The meaning of ‘not well ventilated’ and ‘cemented partially’ is further clarified in the method section, line numbers 243 to 246.

L185: What criteria were used to identify the seriousness of the asthma condition?

Response to the reviewer: We went by the perception of the patients as mentioned during the interview. The number of attacks, their perceived severity and the limitation on activities of daily living were considered while identifying the seriousness of the condition. 

L194-195: What was the evidence to back the statement?

Response to the reviewer: We have removed the statement. 

L209-L230: Remove

Response to the reviewer: The original text got corrupted, but we have added the original text in this section.

The changes have been carried out in the manuscript using track changes. As requested, we are also submitting a version without track changes.

Hoping for a favorable consideration

With regards

Dr Paudel & coauthors

---

## [Editor Report · Decision Letter 1]

25 Aug 2023

Living with bronchial asthma: a qualitative study among patients in a hill village in Nepal

PONE-D-23-12615R1

Dear Dr. Dr Paudel,

We’re pleased to inform you that your manuscript has been judged scientifically suitable for publication and will be formally accepted for publication once it meets all outstanding technical requirements.

Kind regards,

Sumal Nandasena

Academic Editor

PLOS ONE
---

## [Editor Report · Acceptance letter]

1 Sep 2023

PONE-D-23-12615R1 

Living with bronchial asthma: a qualitative study among patients in a hill village in Nepal 

Dear Dr. Paudel:

I'm pleased to inform you that your manuscript has been deemed suitable for publication in PLOS ONE. Congratulations! Your manuscript is now with our production department. 

Kind regards, 

on behalf of

Dr. Sumal Nandasena 

Academic Editor

PLOS ONE